# All-Waste Hybrid Composites with Waste Silicon Photovoltaic Module

**DOI:** 10.3390/polym12010053

**Published:** 2019-12-31

**Authors:** Mihaela Cosnita, Ileana Manciulea, Cristina Cazan

**Affiliations:** Centre Product Design for Sustainable Development, Transilvania University of Brasov, Eroilor 29, 500036 Brasov, Romania; mihaela.cosnita@unitbv.ro (M.C.); i.manciulea@unitbv.ro (I.M.)

**Keywords:** all waste hybrid composite materials, silicon photovoltaic module, recycling, mechanical properties, interfacial adhesion, water stability

## Abstract

Nowadays, global warming, energy issues and environmental concern have forced energy production stakeholders to find new low carbon solutions. Photovoltaic technologies as renewable energy resources represent a competitive way for the transition from conventional fossil fuels towards a renewable energy economy. The highest renewable energy systems (RES) market share is based on silicon photovoltaic (Si-PV). The installed RES have rapidly increased over the last two decades, but, after the end of their service life, they will be disposed of. Therefore, the constant increase of the installed RES has attracted the global concern due to their impact on the environment and, most of all, due to the content of their valuable resources. However, the rational management of RES waste has not been addressed so far. The paper represents an extension of a previous work focused on Si-PV recycling by developing all waste hybrid composites. The extension research conducted in this paper is related to the influence of Si-PV characteristics on the mechanical performances and water stability of the hybrid composites. All waste hybrid composites developed by embedding different Si-PV grain sizes were tested before and after water immersion in terms of mechanical strength, interfacial adhesion, crystallinity and morphology by scanning electron microscopy (SEM) analyses. The results revealed the better performance of such Si-PV composites compared to that of sieved composites even after long term water immersion. Therefore, high-content Si-PV hybrid composites could be developed without Si-PV powder sieving. Further on, all waste hybrid composites could be used as paving slabs, protective barriers for outdoor applications.

## 1. Introduction

Nowadays, worldwide energy issues and environmental pollution lead to the fast development of renewable energy systems (RES). The highest RES market share is based on silicon photovoltaic (Si-PV) systems which convert sun energy into electricity [1]. Generally, the average service life of a photovoltaic (PV) panel is around 25 years, afterwards, they become waste, raising environmental pollution and human health issues [2,3,4]. In this way, significant PV wastes amounts are expected to be loaded around 2025, and by 2038 the amount of such waste type is expected to reach 1.957.099 t, [2].

The European Union (EU) has already pioneered and implemented policy actions to promote the growth of the new PV recycling industry by introducing the directive on the management of Waste of Electrical, Electronic Equipment (WEEE) in 2012. The 2012 WEEE directive includes a specific regulation for photovoltaics recycling (The European Parliament and the Council of the European Union, 2012) [5]. Thus, a minimum collecting rate of PV average weight of 45% was imposed by 2016 and 65% later. Among the different module technologies on the market, wafer-Si modules have always been the dominant technology with a ~90% market share [2].

In technical terms, the research on solar panel recovery is facing many problems. The key issue is to develop economically feasible and non-toxic recycling technologies. The research on the management of solar photovoltaic panels’ management at the end of their service life is just beginning in many countries. However, there is an urgent need for further expansion of producer responsibility [6,7,8].

There are papers reporting on thermal treatments of the PV waste to remove the ethylene-vinyl acetate (EVA) foil but the process is high energy consumption [9]. Others studies followed the organic dissolution of EVA being a long lasting process with the release of large volatile organic amounts [10,11].

Our previous work has reported the results regarding the silicon photovoltaic modules recycling by embedding them in other problematic waste polymeric blends and developing new composite materials [12]. Rubber, polyvinyl chloride (PVC), high-density polyethylene (HDPE) represent other problematic polymer wastes. The fast development of the automotive industry, the increased amount of tires rubber waste with a negative impact on the environment has forced the European Union (EU) to adopt a new directive on the end-of-life of vehicles (ELV). The ELV Directive states that by 2015 vehicles must be made of 95% recyclable materials [13]. Huge tires amounts are yearly discarded with very few recycling rates and they need a very long degradation time because of their cross-linked structure, stabilizers and other additives content [14,15].

However, they are valuable materials which alongside other polymer wastes and inorganic fillers such as Si-PV wastes could be recycled and new value-added materials could be developed. Moreover, the incorporation of inorganic particles in polymer phases is known to enhance the properties of the resulted organic-inorganic composite material. Therefore, the thermal, mechanical, rheological, fire-resisting properties of the final composite will be improved [16]. Silica (SiO_2_), mainly, is widely used in the tire industry to improve abrasion resistance, wet skid resistance and to decrease rolling resistance for the production of the “green automobile tire” with reduced fuel consumption, because of the requirement of environmental protection and resource-saving. Thus, SiO_2_/rubber nanocomposites have attracted considerable interest over the recent decade [17,18].

This paper continues the previous work focusing on the influence of low Si-PV content and grain size on the performance, even in wet environments, of the all waste hybrid composites with waste polyvinyl chloride (PVC), high-density polyethylene (HDPE) and rubber. All these research studies have been conducted for finding sustainable ways of Si-PV recycling further.

## 2. Characterization of the All-Polymer Waste Composites

### 2.1. Mechanical Testing

Tensile strength (*R*_T_) was measured with the Z020, Zwick/Roell equipment (Zwick Roell, Furstenfeld, Austria), at a traction speed of 100 mm/min and Young modulus (*E*) was calculated with the soft equipment. The compressive strength (*R*_C_) was tested on the same mechanical testing equipment, according to SR EN ISO 527-4: 2000 and impact resistance (*R*_I_) were evaluated with the IZOD method based on the SR EN ISO 180: 2001 standard and using the impact test equipment Galdabini Impact (Galdabini, Iserlohn, Germany).

### 2.2. Fourier Transform Infrared Spectroscopy (FTIR) Analysis

The chemical structure changes, especially at the interface zone of all waste hybrid composites, were analysed by recording their infrared (IR) spectra. The Fourier Transform Infrared Spectroscopy (FTIR) analysis was run in reflectance mode, by using a spectrophotometer (Spectrum BX Perkin Elmer, Markham, ON, Canada). The FTIR spectra were recorded in the range 500 to 4500 cm^−1^, after 16 scans, with a 4 cm^−1^ resolution.

### 2.3. X-ray Analysis

The crystalline structure and the influence of the Si-PV particles on the composite’ interface were investigated with the aid of the X-ray diffraction measurements (Bruker D8 advance Diffractometer (Bruker, Ettlingen, Germany) using Cu K_α_ radiation with a wavelength of 1.54016 Å). The data were collected over the range 2θ = 10°–80° with a step interval of 0.01° at 25 °C temperature.

## 3. Experimental Set-Up

### 3.1. Materials

The materials used for processing the all-waste hybrid composites are below. Tire rubber powder with a particle size of 1 mm (which consists of four different types of rubber: natural, styrene-butadiene, polybutadiene and butyl-rubber) from Granutech Recycling (Suceava, Romania). Polyvinyl chloride (PVC) flakes and high-density polyethylene (HDPE) flakes are coming from a collecting and recycling company-Silnef SRL, Brasov, Romania, the PVC derives from shredded and milled cables containing tris (2,4-di-tert-butylphenyl) phosphite as stabilizer agent. Si-PV modules waste (waste from the Research Institute of Transilvania University of Brasov, Brasov, Romania). The PVC and HDPE flakes were milled by using the Centrifugal mill ZM 200 (Retsch, Haan, Germany) to obtain a powder with the diameter of the grain less than 1 mm.

The dismantling of the end-of-life Si-PV modules (poly-crystalline Si-PV modules) was manually performed and the aluminium frame and junction box were removed, Figure 1A, to be sent to specialized recycling companies. Afterwards the Si-PV wafer was chopped in small parts, Figure 1B and milled Figure 1C. The Si-PV powder resulted after milling was added to rubber-HDPE blend. The blend containing rubber-HDPE-Si-PV was compression moulded and thermal processed resulting the rectangular samples shown in Figure 1.

The experimental steps in the development of all waste hybrid composites are shown below in Figure 1.

As already mentioned, this study is an extension of previous research on all waste Si-PV based polymer composites [12]. In the first stage of the previous study on Si-PV based all-polymer waste composites, checks were conducted on the effect of the low Si-PV powder content (between 0.5–3 wt %) with and without glass components, on the interface/mechanical properties of the polymer blends (rubber, PVC and HDPE). The previous study results had indicated the benefit on mechanical performance when inserting all Si-PV, which means composite preparation by inserting the entire milled silicon wafer (including glass). In that study, the Si-PV particles had not been sieved, and the subsequent question was: which is the influence of the Si-PV grain size on the interface and stability of the Si-PV based all-polymer waste composites? That is why this study was conducted on Si-PV grain size influence at composite’ interface level, considering all Si-PV powder (with glass) and the water stability of these composites.

The four sample series were prepared by sieving the Si-PV powder in 20, 40, 100 and 200 microns mesh followed by their insertion in the waste polymer blend (PVC, rubber and HDPE), for the composition ratio of PVC:rubber:HDPE:Si-PV = 60:32:5:3, all composite components in wt %. Each sample series (with 20, 40, 100 and 200 microns Si-PV powder) has been replicated three times. The 1st being tested as such, the 2nd- after 500 h water immersion, the 3rd- after 1000 h water immersion. The mechanical properties of the newly developed composite materials are presented in what follows.

### 3.2. Results and Discussions

#### 3.2.1. Mechanical Properties

The measurements of the mechanical properties are determinants for these kinds of composite materials, as they could be used as products for various applications such as protective barriers, paving slabs, covers for playgrounds and so on. All these applications should meet mechanical properties requirements as to tensile, compressive, impact strength.

The key factors with overwhelming influence on the output properties (mechanical performance) of the composite materials are interfacial bonds, determined in turn by the processing parameters, filler dispersion rate, filler wetting, filler shape and size, components properties, the mass ratio of the components. The larger the accessible surface, the higher the interphase fraction which is expected. It is worth noting that the interactions among filler particles are generally much stronger than interfacial interaction [17,18,19].

The assessment of the behaviour of the material under outdoor conditions is of the utmost importance when outdoor applications are targeted. This study evaluated the mechanical strength of the novel composite materials developed after their water immersion.

Five representative samples of each series were mechanically tested and the average values are summarized in Table 1.

The mechanical characteristics of the etalon sample (without any Si-PV addition) before and after water immersion are presented in Table 1. In the tables below the tensile strength is denoted as *R*_T_, compression strength as *R*_C_ and Young moduli as *E*. All dates are reported with standard deviation (Sd).

By comparing the mechanical properties of the polymeric composites with no Si-PV content (Table 1) with that of the composite materials with low Si-PV content, before and after water immersion, the following conclusions could be drawn:

##### Before Water Immersion

The tensile and compression strength of the un-immersed composites are comparable, 2.40 and 29.36 MPa of no Si-PV content composite–Table 1 and 2.36 and 28.54 MPa for 3% Si-PV composite–see Table 2. The tensile strength of the 2.5wt % Si-PV sample had a significant drop compared to the same property of the etalon sample (without Si-PV filler). This drop could be attributed to the Si-PV partial agglomeration.

However, the Young modulus of 3wt % Si-PV content is almost 70% higher (17.16 MPa) than that of the no Si-PV content composite (10.91 MPa), as it can be seen in Table 1 and Table 2. All dates are reported with standard deviation (Sd).

Significant impact strength increase after the Si-PV powder embedding in the polymeric blend was noticed. This effect is due to the higher inorganic filler stiffness compared to that of the polymeric materials, thus leading to the improvement of the composites’ mechanical and thermal properties. These benefits are connected to the ability of the inorganic filler to absorb the mechanical and thermal energy.

##### After Water Immersion

The immersion in water of the composites with no Si-PV content and with low Si-PV content determined a similar trend regarding the tensile, the compression strength and the Young modulus. All these mechanical properties have been improved after their water immersion. But the most significant increase was recorded during the compression strength tests which showed an over 75% compression strength increase after water immersion. The plasticizer effect of water could explain this remarkable compression strength increase and previous studies confirmed that [20].

The plasticizer water effect was more intense when the samples were immersed for a longer period (for 1000 h compared to 500 h), see Table 1 and Table 2 and Figure 2. In both cases, for both no Si-PV content composites and low Si-PV content composites a slight compression strength increase was noticed from about 50 MPa after 500 h of water immersion to about 53 MPa after 1000 h water immersion. The larger the water amount entering the composite capillary structure, the higher the plasticizing effect of water and the higher the composites structure density.

In what follows the influence of the Si-PV grains size on the mechanical properties of the organic-inorganic composite materials, before and after water immersion, is investigated.

The output properties of 3 wt % Si-PV composite materials were determined in terms of tensile, compressive and impact strength, before and after 500 h and 1000 h water immersion, for all four composite series (with 20, 40, 100 and 200 μm Si-PV), the results were summarized in Table 3. All dates are reported with standard deviation (Sd).

Un-Immersed Samples

By comparing the mechanical performance of the pristine composite series (un-immersed), the following conclusions can be drawn:-the tensile strength values are very close to each another in all four composite series (with 20, 40, 100 and 200 μm Si-PV powder). These values, in turn, are comparable with that of the control sample (with no Si-PV powder content) Table 1,-with reference to the compressive strength, the composite with different Si-PV grain sizes exhibited higher compressive strength compared to the no Si-PV content composites, due to the higher inorganic filler stiffness, [21],-the samples with the best compressive strength are of 100 and 200 μm Si-PV, 49.91 and 45.14 MPa, respectively, as it can be seen in Table 3. These values are 69% and 55% higher than that of the control sample (29.36 MPa), Table 1. This could be explained by the reduced macromolecules polymeric chains mobility in the near vicinity of the inorganic particulates of such size,-100 μm Si-PV sample has recorded Young moduli of 26.40 MPa, which is over 100% larger than the Young moduli of the control sample (10.91 MPa). This result again is explained by the reduced mobility of the polymeric chains. Furthermore, because the smaller Si-PV particulates size (100 microns) compared to that composites containing 200 microns Si-PV has expanded the surface contact of the polymeric chains with the inorganic particles and consequently the mobility of a larger polymer chains density has decreased.

It can be noticed that the addition of sieved (large surface area) Si-PV powder does not improve the interfacial strength of the composite materials based on wastes of rubber, PVC and HDPE. The remarkable improvement noticed was on the compressive strength and Young moduli of the samples with low content of Si-PV powder. This result revealed that the Si-PV powder act at the strength level not at the toughness level. Many researchers reported on the ability of the inorganic compound to improve the polymer composite strength [22,23,24].

The inorganic particles coming in the very vicinity of the macromolecular chains determine their constraints. Therefore, the macromolecular phase mobility is reduced, that is why the samples compression strength increases.

But the expectations were in the interfacial improvement, because once the Si-PV particles grain size is reduced, the specific surface area is extended, which in turn should promote an extension of the interface linkage. However, it could happen that, by decreasing the inorganic particles grain size, the distance between them lowers, which in turn reduces the dispersion degree due to the mutual attraction between the inorganic particles. Finally, the agglomeration effect could occur.

In order to evaluate the behaviour of the composites in wet conditions, the samples were water immersed for 500 and 1000 h. Before the mechanical performance measurements, the samples were dried in the open air in the laboratory.

Water immersion induced the plasticization of the composites structure, consequently increasing their stiffness.

It is also to note that the samples tensile strength after water immersion recorded a slight decrease confirming the peculiar capillary structures of the composite samples.

By comparing the mechanical tests results after 500 and 1000 h water immersion, Figure 3, respectively, in both cases, the best mechanical properties combination was recorded for all Si-PV composite samples (un-sieved Si-PV samples), Table 3. It seems that all waste Si-PV constituents better improve the organic-inorganic composites’ mechanical performance. The benefit of inserting entire and un-sieved waste Si-PV is due to its versatile composition containing both the organic phases and inorganic phases. Among which the ethylene-vinyl acetate (EVA) components have a strong ability to bind the inorganic phase to the organic phase.

Below in Figure 4, the relative absorbed water against the water immersion period is plotted (in hours). The samples with 20, 40 and 200 microns Si-PV grains follow the same allure regarding the water absorption rate with increased water absorption in the case of 40 microns Si-PV composites. The highest water absorption rate was recorded for 100 microns Si-PV composites, as shown in Figure 4A,B, is probably attributed to a lower interface adhesion.

Low water absorption was noticed for 20 and 200 Si-PV microns samples, but the lowest and steady over time water absorption was in the case of 200 Si-PV microns samples, see Figure 4.

The versatile composition of the waste un-sieved Si-PV participates in developing interface linkages with both polymer components and the inorganic constituents from the rubber matrix. These aspects are further investigated by FTIR and XRD analyses in the following sections.

#### 3.2.2. XRD Analysis

The XRD measurements were performed to evaluate the influence of the Si-PV particles on the novel composite interface and to determine the crystalline structure. Firstly, the crystalline degree of the 20, 40, 100 and 200 microns Si-PV powder was measured and that of the composites obtained by inserting different Si-PV powder grain size, being subsequently compared to that without any Si-PV content.

The XRD diffractograms of the four different Si-PV powder grains and their crystalline percentage values can be seen in Figure 5.

The XRD measurement results of the different Si-PV powder grains sizes have revealed the crystalline peaks corresponding to silicon compounds and to the ethylene-vinyl acetate (EVA). EVA the component part of Si-PV module is used to protect the PV active surface from humidity and impurities. The highest crystalline percentage calculated by XRD equipment software associated with the 200 microns sieved Si-PV powder was 63.5% (three times higher than the other ones), as summarized in Figure 5. The 200 microns sieved Si-PV powder recorded one more large and intense EVA copolymer peak at about 2θ = 21.5°, as it can be seen in Figure 5, thus explaining its higher crystallinity percentage (63.5%) compared to the other ones [25,26].

The increased crystallinity of the 200 microns Si-PV composites also confirms the better mechanical properties of the composites obtained by inserting 200 microns sieved Si-PV compared to the other ones, as shown below in Table 4.

The crystallinity of the samples containing the smallest and the largest Si-PV grain size, 20 and 200 microns Si-PV composites, respectively, were further investigated. The crystalline degrees of these composites before and after water immersion are presented in Table 5.

The highest crystalline percentage was recorded for the 200 microns Si-PV composites—40% as such and even after 1000 h water immersion—59.6% respectively as it was expected. It seems that the water entering the capillary structure of the 200 Si-PV composites determines a slight reorganization of the polymer macromolecular chains, thus leading to an increased ordered structure. This result could be well corroborated with the mechanical test results which recorded the highest compression strength of 50.86 MPa for this composites.

Further, the crystalline structure of the un-sieved Si-PV composite was investigated considering its good mechanical performance, as mechanical test results have previous shown. The XRD patterns of the un-sieved Si-PV composites as such, 500 and 1000 h water immersed, are plotted below in Figure 6.

The crystalline percentages of the un-sieved Si-PV composites, as such, after 500 h and 1000 h water immersion were calculated by means of the XRD equipment software and the results are summarized in Figure 6. The XRD measurements results recorded the best crystalline degree in case of un-sieved Si-PV composites, confirming now the mechanical tests results which revealed the highest compression and impact strength in case of un-sieved Si-PV samples, even after water immersion (compression strength of 53.10 MPa, impact strength of 28.86 kJ\m^2^ after 1000 h water immersion). 

The best output properties (mechanical properties) in the case of un-sieved Si PV samples even after water immersion and confirmed by the XRD measurements could be explained by Si-PV powder composition. The un-sieved Si-PV could contain a higher silica amount which highly contributes to the increase of the mechanical and thermal properties of the elastomer polymers due to its ability as a nucleating agent [27,28].

#### 3.2.3. FTIR Analysis

The FTIR spectroscopy was used to investigate the chemical changes or new interfaces that could be developed between the organic-inorganic components, as the Si-PV powder was sieved (granulated). It was expected that the finer Si-PV grain (such as 20 or 40 microns) should extend the interfacial zone, due to the higher surface contact between the Si-PV powder and the organic polymer phases.

It is worth noting some differences between the all waste hybrid composites obtained with un- sieved Si-PV powder and with different Si-PV grain sizes composites. The FTIR spectrum of the un-sieved Si-PV composites showed some differences compared to the other ones as follows, Figure 7:

-at about 2680 cm^−1^ present new band attributed to the vinyl group from EVA, thus confirming the highest amount of EVA in un-sieved Si-PV composites compared to the other ones. EVA has a high affinity to inorganic filler almost with silica [29],-3400 cm^−1^ band intensity increases of un-sieved Si-PV composites assigned to –OH stretching vibration on the silica surface. This confirms the large silica amount in un-sieved Si-PV composite and the possibility to extend the organic-inorganic interface zone almost with EVA and rubber component,-increased intensity of the bands at about 730 and 1025 cm^−1^ corresponding to Si–O, thus proving a higher silica amount compared to the sieved Si-PV composites,-the 3750 cm^−1^ band broadening due to the increased amount of EVA, which interacts with the rubber component, thus extending interfaces in un-sieved Si-PV composites [30],-shoulder of 870 cm^−1^ bands appeared for un-sieved Si-PV composites as well as for the 200 microns Si-PV composites that could suggest a possible interaction of inorganic components of Si-PV module with PVC polymeric components,

Therefore, because of polymer-filler interactions, some polymer macromolecules were linked on the filler surface, reducing the mobility of the polymer segments and thus increasing their mechanical strength, as mechanical test results already showed. The highest mechanical performance was noticed in the case of un-sieved Si-PV composites.

The results mentioned above emphasise the higher un-sieved Si-PV powder potential to develop extended hybrid organic-inorganic interfaces with the other composite components due to its versatile and higher organic-inorganic content. All these aspects recommending the all Si-PV module powder recycling without its sieving which is beneficial considering the energy consumption during the sieving.

#### 3.2.4. SEM Morphology Investigation

The dispersion of Si-PV powder particles in polymer matrices was evaluated by studying the corresponding SEM micrographs images. The fractured surface morphologies of the cross-section of the composite samples were obtained from the tensile test and the morphologies of the 20, 200 microns Si-PV and the un-sieved Si-PV samples, before and after 1000 h water immersion, were analysed. The SEM images of the fractured tensile tests samples are presented below in Figure 8A–C.

The morphology image of the 20 microns Si-PV sample displayed some delamination and Si-PV particles pull out, as it can be seen in Figure 8A. These aspects could be explained by the formation of agglomerates [31] of particles. Moreover, the finer grains size of the Si-PV particles and their high surface energy determine interactions that could occur between the silica particles themselves.

The 200 microns Si-PV composite SEM image, see Figure 8B, displays a better interfacial adhesion compared to those above and the un-sieved Si-PV composite exhibits improved interfacial adhesion, as it can be observed in Figure 8C), the PVC and rubber well embedding the un-sieved Si-PV components. The interfacial adhesion of the un-sieved Si-PV composite is explained by the new hybrid interfaces developed due to versatile Si-PV powder composition containing organic phases alongside inorganic constituents, as already the FTIR results have shown the development of new organic-inorganic interfaces.

### 3.3. High-Content Si-PV All Waste Hybrid Composites

In this section, high-content Si-PV all waste hybrid composites were developed. Taking into account the benefit of the un-sieving of Si-PV powder the high Si-PV content composites were obtained by using un-sieved Si-PV powder.

The un-sieved Si-PV powder content was increased in the rubber-plastic blend. The component composition ratio was PVC:rubber:HDPE:Si-PV = (100 − 37 − *x*):32:5:*x*, where *x* = 10, 20, 30 and 45 wt % Si-PV. Five samples were tested for each mechanical property measurement and the mean value alongside standard deviations are reported in Table 6. By increasing the Si-PV amount in the polymeric blend it is expected an increase of the mechanical strength of the all waste hybrid composites. The Si-PV component will contribute to the extension of the interfacial adhesion in the composite because it contains polymer phases alongside inorganic phases.

Especially the EVA polymer phases of the Si-PV it is expected to develop a better linkage between organic-inorganic phases.

The mechanical test results have shown that by increasing the Si-PV content up to 30 wt % in the polymeric blend the mechanical properties increased but over this amount significant mechanical properties decrease was noticed. The mechanical performance decrease for composites with over 30 wt % Si-PV could be determined by filler-filler interactions because of their high surface energies forming thus clusters which hamper the stress transfer between matrix and filler.

It is worthy to note that the impact resistance test has recorded high values even for 45 wt % Si-PV composites, Table 6, outlining thus the possibility of using such composite materials as products for applications where impact strength are required (protective barriers for example).

The best combination of mechanical properties was noticed for 30 mass percentages Si-PV composite with *R*_T_ of 2.02 MPa, impact strength of 25.06 kJ/m^2^ and R_C_ of 31.90 MPa. 

These results would confirm the possibility of recycling this new type of waste–silicon photovoltaic modules (Si-PV) in novel all waste hybrid composites.

## 4. Conclusions

The paper investigated the influence of the Si-PV grains sizes on the mechanical and water stability of all waste hybrid composites. This study is an extension of a previous work in which waste Si-PV module recycling was the focus.

The Si-PV powder was 20, 40, 100, and 200 microns sieved before its inserting in the waste polymer blend. Good mechanical properties even after water immersion were obtained for 200 microns Si-PV composites. However, when compared to those of un-sieved Si-PV composites the last one recorded the highest mechanical properties and good water stability. The XRD, FTIR, and SEM analyses results confirmed the organic-inorganic interface extending which in turn led to the mechanical strength improvement. The XRD analyses revealed the superior properties of the un-sieved Si-PV composites attributing them the highest crystalline percentage. This result is directly related to the organic and inorganic versatile composition of the Si-PV module powder. The FTIR results confirmed the mechanical performance of the un-sieved Si-PV composites due to the development of new hybrid interfaces.

The mechanical properties of sieved Si-PV hybrid composites decreased due to the reduced distance between inorganic particles and their mutual attraction.

In the second part of this study high Si-PV content (up to 45%) was embedded in the polymer blend. The best combination of mechanical properties was noticed for 30 mass percentages Si-PV composite with R_T_ of 2.02 MPa, impact strength of 25.06 kJ/m^2^ and R_C_ of 31.90 MPa, but the impact strength did not decrease even for 45 mass percentages Si-PV composites. The study confirms the possibility of recycling this new type of waste-silicon photovoltaic modules (Si-PV) in novel all waste hybrid composites without Si-PV powder sieving. 

Further, studies will continue to develop all waste high Si-PV content hybrid composites with mechanical performance even in wet conditions for outdoor applications.

## Figures and Tables

**Figure 1 polymers-12-00053-f001:**
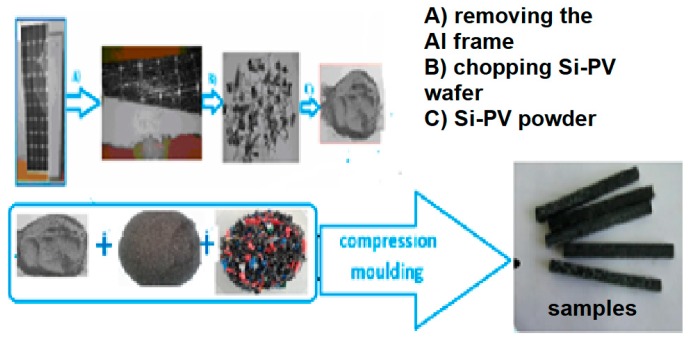
Preparing the waste silicon photovoltaic (Si-PV) powder from PV module and all waste hybrid composites.

**Figure 2 polymers-12-00053-f002:**
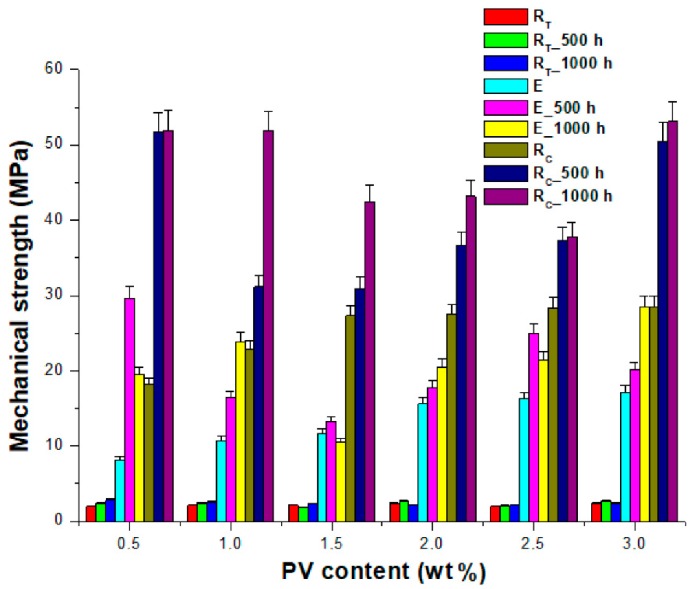
The influence of the Si-PV grains size on the mechanical properties of the hybrid composite materials.

**Figure 3 polymers-12-00053-f003:**
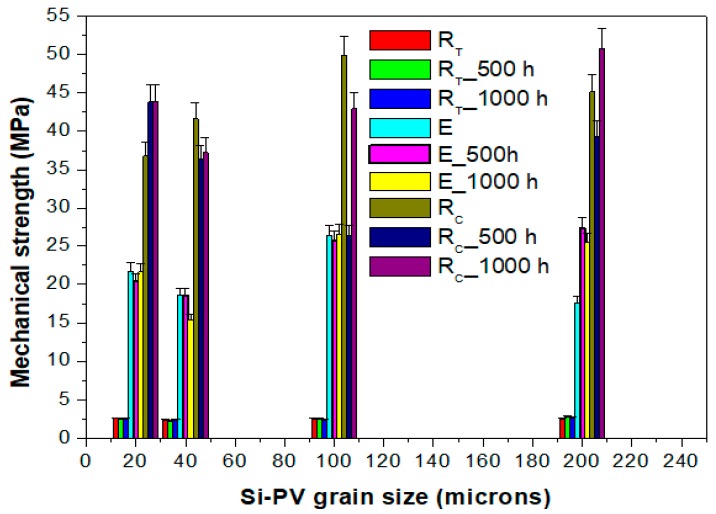
Water immersed samples (500 and 1000 h).

**Figure 4 polymers-12-00053-f004:**
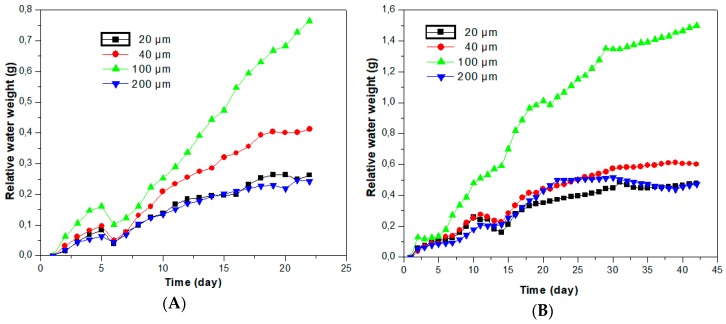
Relative absorbed water after (**A**) 500 h, (**B**) 1000 h.

**Figure 5 polymers-12-00053-f005:**
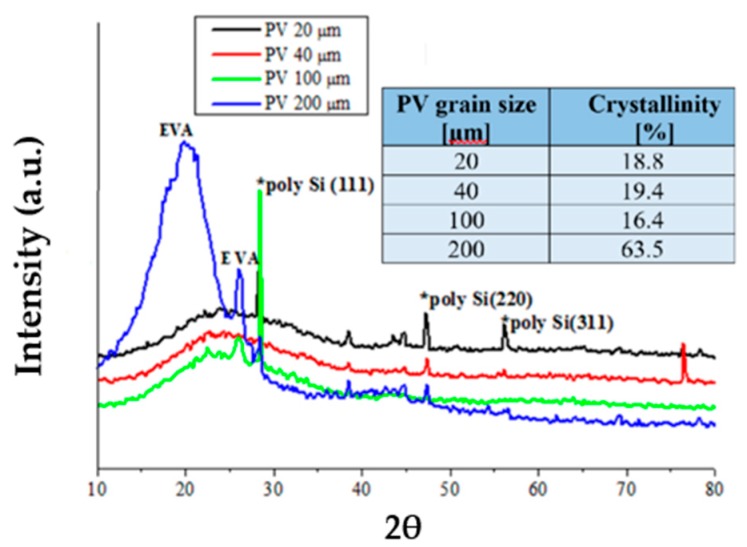
XRD diffractograms of the different Si-PV powder grain.

**Figure 6 polymers-12-00053-f006:**
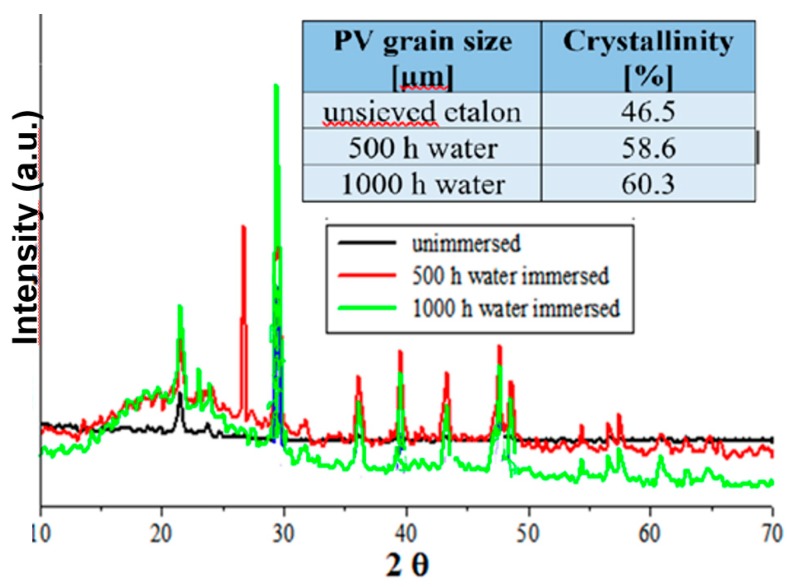
XRD diffractograms of the un-sieved 3% Si-PV hybrid composite.

**Figure 7 polymers-12-00053-f007:**
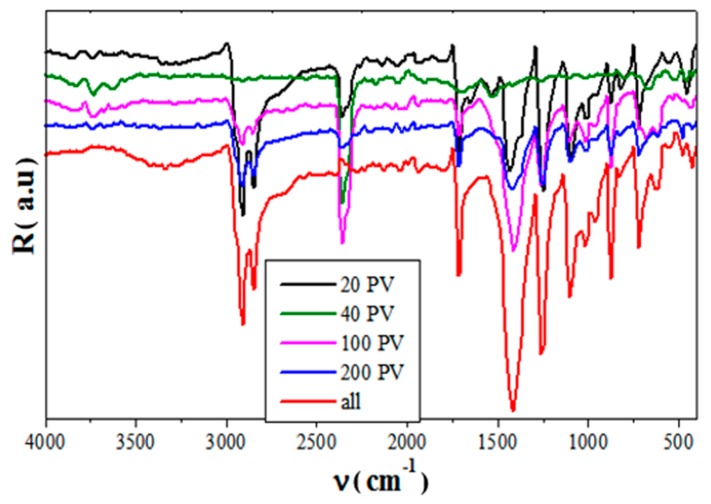
Fourier Transform Infrared Spectroscopy (FTIR) analysis of the 3% PV composite with different Si-PV grain size.

**Figure 8 polymers-12-00053-f008:**
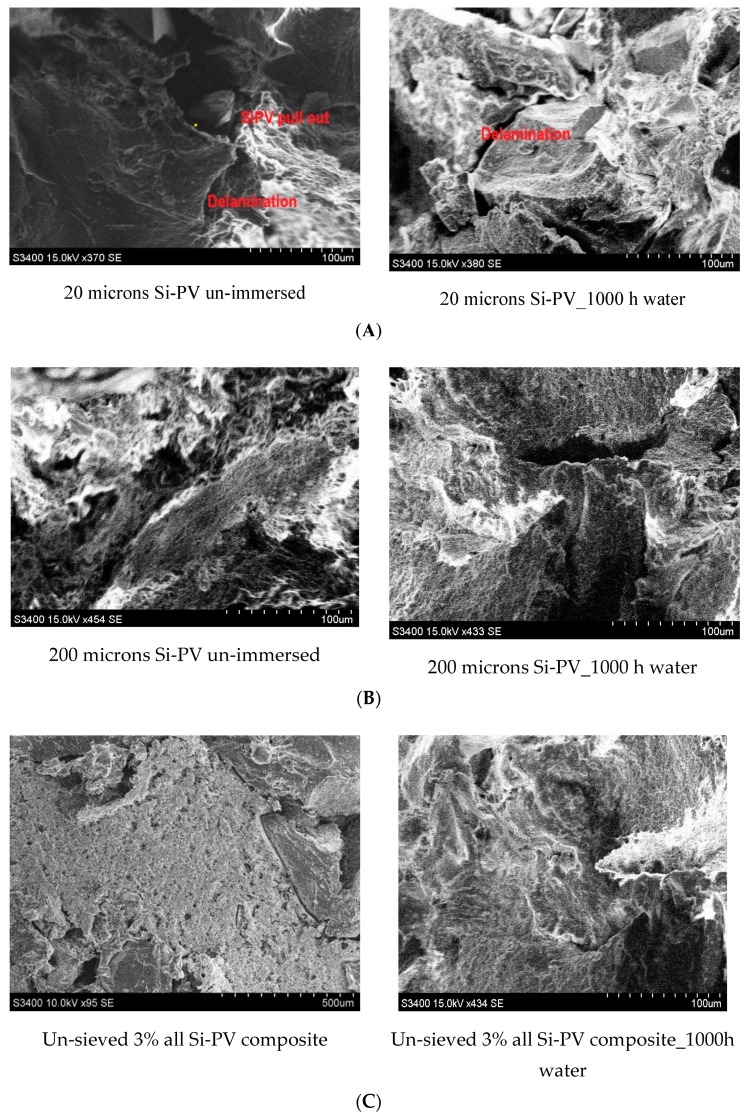
SEM micrographs of: (**A**) 20 microns Si-PV sample, (**B**) 200 microns Si-PV sample, (**C**) un-sieved Si-PV samples.

**Table 1 polymers-12-00053-t001:** Mechanical characteristics of the sample with no Si-PV content.

PVC: Rubber: HDPE: PV	*R*_T_ (MPa)	Sd (for R_T_)	*E* (MPa)	*R*_C_ (MPa)	Sd (for R_T_)	*R*_I_ (kJ/m^2^)	Sd (for R_T_)
60:35:5:0 un-immersed samples	2.4 ± 0.18	0.1848	10.91	29.36 ± 1.16	1.1574	10.72 ± 0.78	0.7800
500 h water immersed	2.62 ± 0.19	0.1873	20.08	50.46 ± 0.90	0.8950	11.2 ± 0.74	0.7371
1000 h water immersed	2.4 ± 0.25	0.2499	28.45	53.1 ± 2.02	2.0154	11.85 ± 0.43	0.4327

**Table 2 polymers-12-00053-t002:** Mechanical properties of all-waste polymer composites with low Si-PV content.

Water Immersion Duration	Si-PV (wt %)	*R*_T_ (MPa)	Sd (for *R*_T_)	*E* (MPa)	*R*_C_ (MPa)	Sd (for *R*_C_)
un-immersed	0.50	1.98 ± 0.19	0.1850	8.18	18.15 ± 0.91	0.9050
	1.00	2.13 ± 0.09	0.0917	10.72	22.85 ± 0.94	0.9425
	1.50	2.13 ± 0.21	0.2113	11.64	27.28 ± 2.04	2.0367
	2.00	2.4 ± 0.40	0.3993	15.62	27.46 ± 2.45	2.4492
	2.50	2.32 ± 0.45	0.4486	15.42	28.29 ± 0.87	0.8712
	3.00	2.39 ± 0.46	0.4572	16.86	28.54 ± 1.01	1.0076
500 h	0.50	2.35 ± 0.40	0.3998	29.64	51.65 ± 0.56	0.5565
	1.00	2.38 ± 0.34	0.3383	16.44	31.14 ± 1.43	1.4312
	1.50	1.82 ± 0.17	0.1701	13.25	30.95 ± 2.00	1.9952
	2.00	2.65 ± 0.26	0.2594	17.74	36.57 ± 1.08	1.0831
	2.50	2.05 ± 0.17	0.1710	24.91	37.24 ± 0.27	0.2651
	3.00	2.62 ± 0.41	0.4105	20.08	50.46 ± 1.03	1.0318
1000 h	0.50	2.76 ± 0.45	0.4521	19.54	51.95 ± 1.02	1.0200
	1.00	2.5 ± 0.45	0.4454	23.83	51.9 ± 1.95	1.9456
	1.50	2.28 ± 0.35	0.3493	10.50	42.48 ± 0.68	0.6843
	2.00	2.12 ± 0.27	0.2730	20.50	43.14 ± 1.41	1.4065
	2.50	2.1 ± 0.30	0.2955	21.42	37.79 ± 0.91	0.9106
	3.00	2.4 ± 0.25	0.2468	28.45	53.1 ± 1.33	1.3268

**Table 3 polymers-12-00053-t003:** Mechanical properties of hybrid composites with low Si-PV content.

PVC: Rubber: HDPE: PV 60:32:5:3	PV Grain Size (μm)	*R*_T_ (MPa)	Sd (for *R*_T_)	*E* (MPa)	*R*_C_ (MPa)	Sd (for *R*_C_)	*R*_I_ (kJ/m^2^)	Sd (for *R*_I_)
un-immersed	20	2.51 ± 0.20	0.1955	21.75	36.75 ± 0.84	0.8351	10.24 ± 0.97	0.9729
	40	2.33 ± 0.35	0.3477	18.61	41.62 ± 1.19	1.1856	11.43 ± 0.31	0.3050
	100	2.45 ± 0.23	0.2301	26.40	49.91 ± 0.97	0.9711	11.84 ± 0.87	0.8651
	200	2.44 ± 0.22	0.2194	17.55	45.14 ± 0.06	0.0557	13.54 ± 0.88	0.8827
	all	2.36 ± 0.18	0.1758	17.16	49.54 ± 0.88	0.8780	12.89 ± 0.76	0.7601
500 h water immersed Samples	20	2.4 ± 0.48	0.4800	20.41	43.82 ± 1.21	1.2123	23.59 ± 0.33	0.3332
	40	2.13 ± 0.42	0.4223	18.58	36.34 ± 1.05	1.0512	23.25 ± 0.46	0.4565
	100	2.42 ± 0.24	0.2390	25.72	26.39 ± 1.36	1.3563	23.25 ± 0.56	0.5631
	200	2.66 ± 0.29	0.2851	27.33	39.4 ± 0.25	0.2498	23.59 ± 0.36	0.3568
	all	2.62 ± 0.46	0.4613	20.08	50.46 ± 0.85	0.8455	27.14 ± 0.18	0.1779
1000 h water immersed Samples	20	2.45 ± 0.45	0.4493	21.67	43.94 ± 1.73	1.7266	26.62 ± 0.53	0.5250
	40	2.26 ± 0.32	0.3205	15.34	37.27 ± 0.90	0.8990	23.25 ± 0.57	0.5661
	100	2.26 ± 0.41	0.4062	26.51	42.94 ± 0.24	0.2386	23.25 ± 0.38	0.3751
	200	2.58 ± 0.55	0.5468	25.44	50.86 ± 0.61	0.6067	23.92 ± 0.78	0.7842
	all	2.4 ± 0.37	0.3736	28.45	53.1 ± 0.10	0.0950	28.86 ± 0.87	0.8652

**Table 4 polymers-12-00053-t004:** XRD pattern of Si-PV polymer composites with different Si-PV grain size.

PV Grain Size	No Si-PV	Un Sieved	20 (μm)	40 (μm)	100 (μm)	200 (μm)
**Crystallinity (%)**	63.3	46.5	43.6	47.8	51.5	64.2

**Table 5 polymers-12-00053-t005:** The crystallinity of the 20 and 200 microns Si-PV composites.

PV Grain Size (μm)	Crystallinity (%)
20_un-immersed	38.9
20_500 h water immersed	38.8
20_1000 h water immersed	48.3
200_ un-immersed	40.0
200_500 h water immersed	43.5
200_1000 h water immersed	59.6

**Table 6 polymers-12-00053-t006:** Mechanical tests results of increased all Si-PV content composites.

Si-PV (wt %)	*R*_T_ (MPa)	Sd (for *R*_T_)	*E* (MPa)	*R*_C_ (MPa)	Sd (for *R*_C_)	*R*_I_ (kJ/m^2^)	Sd (for *R*_I_)
10	1.92 ± 0.21	0.2117	12.25	48.63 ± 0.48	0.4770	23.25 ± 0.34	0.3395
20	1.25 ± 0.08	0.0781	5.78	37.74 ± 0.45	0.4479	22.86 ± 0.26	0.2566
30	2.02 ± 0.09	0.0872	2.84	30.81 ± 0.51	0.5069	25.06 ± 0.41	0.4095
40	0.63 ± 0.02	0.0153	0	34.34 ± 0.29	0.2862	23.42 ± 0.56	0.5565
45	0.5 ± 0.04	0.0361	0	32.99 ± 0.32	0.3219	23.58 ± 0.52	0.5179

## Data Availability

The raw/processed data required to reproduce these findings cannot be shared at this time as the data also forms part of an ongoing study.

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
