# Peer review of "All-Waste Hybrid Composites with Waste Silicon Photovoltaic Module"

_polymers, 2019, doi:10.3390/polym12010053_

Round 1

Reviewer 1 Report

Again "In Materials section Authors should provide more information regarding used materials, some characteristics, trade names, etc.". On the market there are at least hundreds of types of HDPE or PVC. Did PVC contain any additives? It is important information. With given information it is totally not replicable.

"Q2. Why Authors applied only up to 3 wt% of waste filler? If it is a waste then it is beneficial to introduce higher amount and check where is the optimum connecting high filler content with satisfactory mechanical performance.

A2. Thank you this is a good question! You are right, but this study represent the first part of ongoing research. First of all, the focus was on checking if this kind of waste (SiPV) could develop interfaces with rubber-plastic blend and consequently if could be recycled in all waste hybrid composites."

For me it is rather poor answer. Such combination of PVC, rubber and HDPE would easily consume such small amounts. It can be clearly seen that considering standard deviations hardly any changes were observed in mechanical performance. It the aim is to recycle this type of waste by manufacturing of composites then higher contents have to be investigated, because at such low contents it makes hardly any sense. 

Regarding standard deviation values, if it reaches or exceeds 20% I believe that some tests have to be repeated to eliminate extreme results.

"Q4. What is the reason for noticeable drop of mechanical performance of unimmersed samples with 2.5 wt% of filler? Same for compression strength for 3 wt% after previous drop for lower contents?
Same drop for particle size of 40 µm, why?

A4. The mechanical performance of un-immersed 2.5 % Si-PV composite are: tensile strength (RTof 1.98 MPa, compression strength of 28.29 MPa and Young moduli of 16.31 MPa (Table 2) compared to the etalon (without Si-PV addition) with tensile strength of 2.40, compression of 29.36 MPa and the young moduli of 10.91 MPa (Table 1). The tensile strength of 2.5 % Si-PV sample is the property with a significant drop compared to the same property of the etalon sample (without Si-PV filler). This drop could be attributed to the Si-PV partial agglomeration. These explanations was added in the manuscript in the 3.2.1 section with red letters.

On the other hand, the compression strength of the 3 % Si-PV sample (28.54 MPa) as can be seen in Table 2 is slightly lower than that of the etalon sample (29.36 MPa), Table 1. In addition the Young moduli of the 3 % Si-PV sample (17.16 MPa) is the highest in the un-immersed and un-sieved samples series.

The mechanical properties of the 40 microns Si-PV sample are better than that of the etalon sample excepting a slight drop in the case of the tensile strength (Table 3). The tensile strength is 2.33 MPa, compression of 41.62 MPa (much larger than that of the etalon) and the young moduli of 18.61 MPa."

Ok, in the response Authors mentioned properties given in Tables, I can also check them. But I still do not know why agglomeration was observed for 2.5% content and not for 3%? Why such irregularity was observed? Because as I see it samples with 2.5% are just defected and should be prepared again. 

Author Response

Point 1: Again "In Materials section Authors should provide more information regarding used materials, some characteristics, trade names, etc.". On the market there are at least hundreds of types of HDPE or PVC. Did PVC contain any additives? It is important information. With given information it is totally not replicable.

Response 1: The waste materials (HDPE and PVC) used are coming from a collecting and recycling company- Silnef SRL, Brasov, Romania. The PVC derives from shredded and milled cables containing tris (2,4-di-tert-butylphenyl) phosphite as stabilizer agent Tris(2,4-di-tert-butylphenyl)phosphite.

These informations were added with red letters in the manuscris at section 3.1 Materials.

Point 2: For me it is rather poor answer. Such combination of PVC, rubber and HDPE would easily consume such small amounts. It can be clearly seen that considering standard deviations hardly any changes were observed in mechanical performance. It the aim is to recycle this type of waste by manufacturing of composites then higher contents have to be investigated, because at such low contents it makes hardly any sense. 

Response 2: The Si-PV powder content was increased up to 45 wt% in the rubber-plastic blend. The component composition ratio was PVC: rubber: HDPE: Si-PV= (100-37-x): 32: 5: x, where X= 10, 20, 30, 40 and 45 % Si-PV.  Five samples were tested for each mechanical property measurement and the mean value alongside standard deviations are reported in Table 6.

Table 6 Mechanical tests results of increased all Si-PV content composites

Si-PV   

[wt %]

RT

[MPa]

Sd

(for RT)

E

[MPa]

RC

[MPa]

Sd

(for RC)

RI

(kJ/m2)

Sd

(for RI)

10

1.92±0.21

0.2117

12.25

48.63±0.48

0.4770

23.25±0.34

0.3395

20

1.25±0.08

0.0781

5.78

37.74±0.45

0.4479

22.86±0.26

0.2566

30

2.02±0.09

0.0872

2.84

30.81±0.51

0.5069

25.06±0.41

0.4095

40

0.63±0.02

0.0153

0

34.34±0.29

0.2862

23.42±0.56

0.5565

45

0.5±0.04

0.0361

0

32.99±0.32

0.3219

23.58±0.52

0.5179

By increasing the Si-PV amount in the polymeric blend it is expected an increase of the mechanical strength of the all waste hybrid composites. The Si-PV component will contribute to the extension of the interfacial adhesion in the composite because it contains polymer phases alongside inorganic phases. Especially the EVA polymer phases of the Si-PV it is expected to develop a better linkage between organic-inorganic phases. The mechanical test results have shown that by increasing the Si-PV content up to 30% wt in the polymeric blend the mechanical properties increased but over this amount significant mechanical properties decrease was noticed. The mechanical performance decrease for composites with over 30 % wt Si-PV could be determined by filler-filler interactions because of their high surface energies forming thus clusters that hamper the stress transfer between matrix and filler. It is worthy to note that the impact resistance test has recorded high values even for 45 % wt Si-PV composites, Table 6, outlining thus the possibility of using such composite materials as products for applications where impact strength is required (protective barriers for example). The best combination of mechanical properties was noticed for 30 % wt Si-PV composite with RT of 2.02 MPa, impact strength of 25.06 kJ/m2 and RC of 31.90 MPa.

These results would confirm the possibility of recycling this new type of waste – silicon photovoltaic modules (Si-PV) in novel all waste hybrid composites.

All these were added in the manuscript with red letters.

Point 3: Regarding standard deviation values, if it reaches or exceeds 20% I believe that some tests have to be repeated to eliminate extreme results.

Response 3: For the samples with 0.5 % Si-PV water immersed for 500 h and 1000 h were repeated the mechanically testing and the results are presented in Table 2 and Table 3 with red letters.

 Point 4: Ok, in the response Authors mentioned properties given in Tables, I can also check them. But I still do not know why agglomeration was observed for 2.5% content and not for 3%? Why such irregularity was observed? Because as I see it samples with 2.5% are just defected and should be prepared again. 

Response 4: The other two series of samples were prepared with 2.5 and 3% Si-PV and the results are reported in Table 2 with red letters in the manuscript.

Table 2. Mechanical properties of all- waste polymer composites with low Si-PV

Water

immersion

duration

Si-PV   

[wt %]

RT

[MPa]

Sd

(for RT)

E

[MPa]

RC

[MPa]

Sd

(for RC)

un-immersed

0.50

1.98±0.19

0.1850

8.18

18.15±0.91

0.9050

1.00

2.13±0.09

0.0917

10.72

22.85±0.94

0.9425

1.50

2.13±0.21

0.2113

11.64

27.28±2.04

2.0367

2.00

2.4±0.40

0.3993

15.62

27.46±2.45

2.4492

2.50

2.32±0.45

0.4486

15.42

28.29±0.87

0.8712

3.00

2.39±0.46

0.4572

16.86

28.54±1.01

1.0076

Reviewer 2 Report

The data in Tables 1-3 should be supplied in the form A±B considering the statistical scattering of the experimental results as it is accepted in the scientific literature.   

Author Response

Point 1: The data in Tables 1-3 should be supplied in the form A±B considering the statistical scattering of the experimental results as it is accepted in the scientific literature.   

Response 1: The data were supplied in the manuscript in the form of A±B as follows:

Table 1. Mechanical characteristics of the sample with no Si-PV content.

PVC: rubber: HDPE: PV

RT

[MPa]

Sd

(for RT)

E

[MPa]

RC

[MPa]

Sd

(for RT)

RI

[kJ/m²]

Sd

(for RT)

60:35:5:0

un-immersed samples

2.4±0.18

0.1848

10.91

29.36±1.16

1.1574

10.72±0.78

0.7800

500 h water immersed

2.62±0.19

0.1873

20.08

50.46±0.90

0.8950

11.2±0.74

0.7371

1000 h water immersed

2.4±0.25

0.2499

28.45

53.1±2.02

2.0154

11.85±0.43

0.4327

Table 2. Mechanical properties of all- waste polymer composites with low Si-PV content.

Water

immersion

duration

Si-PV   

[wt %]

RT

[MPa]

Sd

(for RT)

E

[MPa]

RC

[MPa]

Sd

(for RC)

un-immersed

0.50

1.98±0.19

0.1850

8.18

18.15±0.91

0.9050

1.00

2.13±0.09

0.0917

10.72

22.85±0.94

0.9425

1.50

2.13±0.21

0.2113

11.64

27.28±2.04

2.0367

2.00

2.4±0.40

0.3993

15.62

27.46±2.45

2.4492

2.50

2.32±0.45

0.4486

15.42

28.29±0.87

0.8712

3.00

2.39±0.46

0.4572

16.86

28.54±1.01

1.0076

500 h

0.50

2.35±0.40

0.3998

29.64

51.65±0.56

0.5565

1.00

2.38±0.34

0.3383

16.44

31.14±1.43

1.4312

1.50

1.82±0.17

0.1701

13.25

30.95±2.00

1.9952

2.00

2.65±0.26

0.2594

17.74

36.57±1.08

1.0831

2.50

2.05±0.17

0.1710

24.91

37.24±0.27

0.2651

3.00

2.62±0.41

0.4105

20.08

50.46±1.03

1.0318

1000 h

0.50

2.76±0.45

0.4521

19.54

51.95±1.02

1.0200

1.00

2.5±0.45

0.4454

23.83

51.9±1.95

1.9456

1.50

2.28±0.35

0.3493

10.50

42.48±0.68

0.6843

2.00

2.12±0.27

0.2730

20.50

43.14±1.41

1.4065

2.50

2.1±0.30

0.2955

21.42

37.79±0.91

0.9106

3.00

2.4±0.25

0.2468

28.45

53.1±1.33

1.3268

Table 3. Mechanical properties of hybrid composites with low Si-PV content.

PVC: rubber: HDPE: PV

60:32:5:3

PV grain size [μm]

RT

[MPa]

Sd

(for RT)

E

[MPa]

RC

[MPa]

Sd

(for RC)

RI

[kJ/m²]

Sd

(for RI)

un-immersed

20

2.51±0.20

0.1955

21.75

36.75±0.84

0.8351

10.24±0.97

0.9729

40

2.33±0.35

0.3477

18.61

41.62±1.19

1.1856

11.43±0.31

0.3050

100

2.45±0.23

0.2301

26.40

49.91±0.97

0.9711

11.84±0.87

0.8651

200

2.44±0.22

0.2194

17.55

45.14±0.06

0.0557

13.54±0.88

0.8827

all

2.36±0.18

0.1758

17.16

49.54±0.88

0.8780

12.89±0.76

0.7601

500 h water immersed

Samples

20

2.4±0.48

0.4800

20.41

43.82±1.21

1.2123

23.59±0.33

0.3332

40

2.13±0.42

0.4223

18.58

36.34±1.05

1.0512

23.25±0.46

0.4565

100

2.42±0.24

0.2390

25.72

26.39±1.36

1.3563

23.25±0.56

0.5631

200

2.66±0.29

0.2851

27.33

39.4±0.25

0.2498

23.59±0.36

0.3568

all

2.62±0.46

0.4613

20.08

50.46±0.85

0.8455

27.14±0.18

0.1779

1000 h water immersed

Samples

20

2.45±0.45

0.4493

21.67

43.94±1.73

1.7266

26.62±0.53

0.5250

40

2.26±0.32

0.3205

15.34

37.27±0.90

0.8990

23.25±0.57

0.5661

100

2.26±0.41

0.4062

26.51

42.94±0.24

0.2386

23.25±0.38

0.3751

200

2.58±0.55

0.5468

25.44

50.86±0.61

0.6067

23.92±0.78

0.7842

all

2.4±0.37

0.3736

28.45

53.1±0.10

0.0950

28.86±0.87

0.8652

Table 6 Mechanical tests results of increased all Si-PV content composites

Si-PV   

[wt %]

RT

[MPa]

Sd

(for RT)

E

[MPa]

RC

[MPa]

Sd

(for RC)

RI

(kJ/m2)

Sd

(for RI)

10

1.92±0.21

0.2117

12.25

48.63±0.48

0.4770

23.25±0.34

0.3395

20

1.25±0.08

0.0781

5.78

37.74±0.45

0.4479

22.86±0.26

0.2566

30

2.02±0.09

0.0872

2.84

30.81±0.51

0.5069

25.06±0.41

0.4095

40

0.63±0.02

0.0153

0

34.34±0.29

0.2862

23.42±0.56

0.5565

45

0.5±0.04

0.0361

0

32.99±0.32

0.3219

23.58±0.52

0.5179

Round 2

Reviewer 1 Report

Ok

This manuscript is a resubmission of an earlier submission. The following is a list of the peer review reports and author responses from that submission.

Round 1

Reviewer 1 Report

The present paper by Cazan and co-worker describes the preparation of hybrid composites containing Si photovoltaic wastes. Unfortunately, I do not think that the ms can be published at present.

First of all, the style is really verbose and the ms should be entirely rewritten. Also, I find that a clear explanation of the possible application of the prepared samples is completely missing. Why is it so important to determine the mechanical properties of these materials?

Second, I think that all the measurements should be accompanied by an indication of the associated uncertainty (e.g. standard deviation). As an example, I am skeptical about the difference between parts A and B of Figure 4, since they should offer the same trend up to 22 days, but this is not the case. Furthermore, the same holds true for Tables 4 and 5. Also, it is not clear to me the tags "20 unsieved" and "200 unsieved". Is the material sieved or not? Please comment on the possible differences among samples.
Please notice that I don't consider acceptable the "data availability" sentence. If the Authors do not consider the presented data a complete set, then they should finalize the investigation and then submit the full work when ready.

Finally, please define EVA and all the acronyms in the text.

Author Response

Thank you for all your comments and suggestions.

 Q1. First of all, the style is really verbose and the ms should be entirely rewritten

A1. All manuscript was checked by a native English speaker and all changes over the manuscript are marked with red letters.

Q2. Also, I find that a clear explanation of the possible application of the prepared samples is completely missing. Why is it so important to determine the mechanical properties of these materials?

A2. These kinds of composite materials could be used mainly as products for various applications such as, paving slabs, covers for playgrounds, protective barriers and so on. That why mechanical properties are important. All these applications require mechanical as tensile, compressive, impact strength and so on.

This specification was added with red letters in 3.2.1 Section in the manuscript.

Q3. Second, I think that all the measurements should be accompanied by an indication of the associated uncertainty (e.g. standard deviation).

A3. Thank you for the suggestion. The standard deviation was added for the measurements.

Q4. Also, it is not clear to me the tags "20 unsieved" and "200 unsieved". Is the material sieved or not? Please comment on the possible differences among samples.

A4. The SiPV powder resulted from silicon photovoltaic wafer shredding and milling was sieved in 20, 40, 100 and 200 microns mesh. As was mentioned (page 3, lines 121-127) the SiPV powder sieving was performed in order to study the influence of the SiPV particle size on the interface and mechanical properties of the all waste SiPV based composites. Four samples series were prepared with different SiPV grain sizes: 20, 40, 100 and 200 microns respectively and a reference one with un sieved SiPV powder denoted in the manuscript as all SiPV.

Q5. Finally, please define EVA and all the acronyms in the text.

A5. The EVA – ethylene vinyl acetate respectively was defined in the manuscript alongside all the acronyms. All additions defining the acronyms are marked in the manuscript with red letters.

Reviewer 2 Report

In Materials section Authors should provide more information regarding used materials, some characteristics, trade names, etc. In case of tire rubber particle size should be mentioned. 

Why Authors applied only up to 3 wt% of waste filler? If it is a waste then it is beneficial to introduce higher amount and check where is the optimum connecting high filler content with satisfactory mechanical performance.

Values of mechanical parameters definitely have to be presented with standard deviation.

What is the reason for noticeable drop of mechanical performance of unimmersed samples with 2.5 wt% of filler? Same for compression strength for 3 wt% after previous drop for lower contents?

Same drop for particle size of 40 µm, why?

Why EVA in XRD results, when it was not mentioned in materials section?

Why nanoparticles, when their size is 200 microns? 

Author Response

Comments and Suggestions for Authors

Q1. In Materials section Authors should provide more information regarding used materials, some characteristics, trade names, etc. In case of tire rubber particle size should be mentioned. 

Thank you for your comments and suggestions.

A1. The requested information was provided in the manuscript with red letters. The tire rubber particle size was 1 mm.

                                                  Q2. Why Authors applied only up to 3 wt% of waste filler? If it is a waste then it is beneficial to introduce higher amount and check where is the optimum connecting high filler content with satisfactory mechanical performance.

A2. Thank you this is a good question! You are right, but this study represent the first part of ongoing research. First of all, the focus was on checking if this kind of waste (SiPV) could develop interfaces with rubber-plastic blend and consequently if could be recycled in all waste hybrid composites.

Q3. Values of mechanical parameters definitely have to be presented with standard deviation.

A3. The standard deviation was added for the mechanical parameters.

Q4. What is the reason for noticeable drop of mechanical performance of unimmersed samples with 2.5 wt% of filler? Same for compression strength for 3 wt% after previous drop for lower contents?
Same drop for particle size of 40 µm, why?

A4. The mechanical performance of un-immersed 2.5 % Si-PV composite are: tensile strength (RT of 1.98 MPa, compression strength of 28.29 MPa and Young moduli of 16.31 MPa (Table 2) compared to the etalon (without Si-PV addition) with tensile strength of 2.40, compression of 29.36 MPa and the young moduli of 10.91 MPa (Table 1). The tensile strength of 2.5 % Si-PV sample is the property with a significant drop compared to the same property of the etalon sample (without Si-PV filler). This drop could be attributed to the Si-PV partial agglomeration. These explanations was added in the manuscript in the 3.2.1 section with red letters.

On the other hand, the compression strength of the 3 % Si-PV sample (28.54 MPa) as can be seen in Table 2 is slightly lower than that of the etalon sample (29.36 MPa), Table 1. In addition the Young moduli of the 3 % Si-PV sample (17.16 MPa) is the highest in the un-immersed and un-sieved samples series.

The mechanical properties of the 40 microns Si-PV sample are better than that of the etalon sample excepting a slight drop in the case of the tensile strength (Table 3). The tensile strength is 2.33 MPa, compression of 41.62 MPa (much larger than that of the etalon) and the young moduli of 18.61 MPa

Q5. Why EVA in XRD results, when it was not mentioned in materials section?

A5. Ethylene vinyl acetate (EVA) is a component part of the SiPV module. This information was introduced in the text with red letters. The EVA is the material used to protect the PV active surface from humidity and impurities and raises difficulties in recycling silicon PV panels.

Q6. Why nanoparticles, when their size is 200 microns? 

A6. Thank you for the revision you are right they are not nanoparticles. The correction was applied over the all manuscript by replacing nanoparticles with particles. These changes are marked in the manuscript with red letters.

Reviewer 3 Report

The paper is devoted to the important theme. However it needs a very serious revision.

All of the reported quantitative data reported in the paper and supplied in the tables does not present the statistical scattering of the experimental results, which is well expected to be high. This is a very bad methodological mistake and it definitely should be corrected under the revision.  in the text: "But when compared to those of un-sieved Si-PV composites the last one has recorded the highest mechanical properties and good water stability as the XRD, FTIR and SEM analyses results have confirmed". Actually, XRD, FTIR and SEM analysis say nothing about the mechanical properties of the material. This is a misleading erroneous statement. Figure 1 needs the scale bar.

Author Response

Thank you for your comments.

Point 1: All of the reported quantitative data reported in the paper and supplied in the tables does not present the statistical scattering of the experimental results, which is well expected to be high. This is a very bad methodological mistake and it definitely should be corrected under the revision.

Response 1: The soft of the mechanical testing equipment calculates  the average of the all measurements automatically; that is why it is impossible to provide the statistical scattering.

Point 2: n the text: "But when compared to those of un-sieved Si-PV composites the last one has recorded the highest mechanical properties and good water stability as the XRD, FTIR and SEM analyses results have confirmed". Actually, XRD, FTIR and SEM analysis say nothing about the mechanical properties of the material. This is a misleading erroneous statement.

Response 2: Yes you are right, in fact the  XRD, FTIR and SEM analyses results have confirmed the organic-inorganic interface extending which in turn led to the mechanical strength improvement. This explanation have been added in the manuscripts and marked with red bars.

Point 3: Figure 1 needs the scale bar.

Response 3: Figure 1 actually represents a graphical abstract of this study.

Round 2

Reviewer 1 Report

The revised version of the ms has not improved with respect to the original version, where some comments were analogous to those of other reviewers. In particular:

language and style still need to be improved; uncertainties still need to be added to the tables; Still, I don't understand how the Authors can associate a "grain size" with the label "unsieved". The definition of the grain size requires the sample to be sieved in advance, otherwise I assume that a given grain size distribution will result. Accordingly, I recommend to check the "20 unsieved" and "200 unsieved" labels.

In conclusion, I didn't change my opinion with respect to the present version of the ms and I confirm my original recommendation.

Author Response

Point 1: Language and style still need to be improved; uncertainties still need to be added to the tables;

Response 1: The language was once again improved and marked with red letters.

Point 2:  Accordingly, I recommend to check the "20 unsieved" and "200 unsieved" labels.

Response 2: In Table 5 there was a mistake, actually it was un-immersed not unsieved. the mistakes were corrected and marked with red bar.

Reviewer 2 Report

In previous review I asked about standard deviation, it should be given in tables, because I believe that Authors made at least 3 tests for each specimen.

And about agglomeration of particles for 2.5 wt%, why it is observed for this content and not for higher content of 3.0 wt% where mechanical properties are rising again?

Author Response

Point 1: In previous review I asked about standard deviation, it should be given in tables, because I believe that Authors made at least 3 tests for each specimen

Response 1:The soft of the mechanical testing equipment calculates the average of the all measurements automatically; that is why it is impossible to provide the standard deviation in tables.

Point 2: About agglomeration of particles for 2.5 wt%, why it is observed for this content and not for higher content of 3.0 wt% where mechanical properties are rising again?

Response 2: This behavior in mechanical properties is due to the versatile organic-anorganic composition of the Si-PV waste powder. The 3.0 wt% Si-PV sample seems to be richer in EVA content which extends the organic-inorganic interface (between EVA, rubber and silica). This explanation is well supported by FTIR and XRD analyses.

Reviewer 3 Report

The paper should be rejected.

The authors replied:

Response 1: The soft of the mechanical testing equipment calculates the average of the all measurements automatically; that is why it is impossible to provide the statistical scattering.

I do not accept this reply. Without supplying the standart deviation (the statistical scattering) of the experimental data no paper should be published in the engineering science. I do not understand , what does it mean: "automatic measurements". This is the  misleading, erroneous methodological approach.